# Identification of an Upper Limit of Tumor Burden for Downstaging in Candidates with Hepatocellular Cancer Waiting for Liver Transplantation: A West–East Collaborative Effort

**DOI:** 10.3390/cancers12020452

**Published:** 2020-02-14

**Authors:** Quirino Lai, Alessandro Vitale, Karim Halazun, Samuele Iesari, André Viveiros, Prashant Bhangui, Gianluca Mennini, Tiffany Wong, Shinji Uemoto, Chih-Che Lin, Jens Mittler, Toru Ikegami, Yang Zhe, Shu-Sen Zheng, Yuji Soejima, Maria Hoppe-Lotichius, Chao-Long Chen, Toshimi Kaido, Chung Mau Lo, Massimo Rossi, Arvinder Singh Soin, Armin Finkenstedt, Jean C. Emond, Umberto Cillo, Jan Lerut

**Affiliations:** 1Institut de Recherche Expérimentale et Clinique Université catholique de Louvain, Brussels 1200, Belgium; samuele.iesari@uclouvain.be; 2Hepatobiliary and Organ Transplantation Unit, Sapienza University, Rome 00161, Italy; gianluca.mennini@uniroma1.it (G.M.); massimo.rossi@uniroma1.it (M.R.); 3Hepatobiliary Surgery and Liver Transplant Unit, University of Padua, Padua 35121, Italy; alessandro.vitale@unipd.it (A.V.); cillo@unipd.it (U.C.); 4Columbia University and New York-Presbyterian/Weill Cornell Medical Center, New York, NY 10065, USA; kah7007@med.cornell.edu (K.H.); je111@columbia.edu (J.C.E.); 5Department of Medicine I, Medical University of Innsbruck, Innsbruck 6020, Austria; andre.viveiros@tirol-kliniken.at (A.V.); armin.finkenstedt@tirol-kliniken.at (A.F.); 6Medicine Medanta-The Medicity, Guragram, Dehli 122006, India; pbhangui@gmail.com (P.B.); absoin@gmail.com (A.S.S.); 7Department of Surgery, The University of Hong Kong, Hong Kong, China; wongtcl@hku.hk (T.W.); locm@hku.hk (C.M.L.); 8Division of Hepato-Biliary-Pancreatic and Transplant Surgery, Department of Surgery, Graduate School of Medicine, Kyoto 615-8530, Japan; uemoto@kuhp.kyoto-u.ac.jp (S.U.); kaido@kuhp.kyoto-u.ac.jp (T.K.); 9Chang Gung Memorial Hospital, Kaohsiung 33305, Taiwan; immunologylin@gmail.com (C.-C.L.); clchen@cgmh.org.tw (C.-L.C.); 10Klinik für Allgemein-, Viszeral- und Transplantationschirurgie Universitätsmedizin Mainz, Mainz 76726, Germany; jens.mittler@unimedizin-mainz.de (J.M.); maria.hoppe-lotichius@unimedizin-mainz.de (M.H.-L.); 11Department of Surgery and Science, Kyushu University, Fukuoka 819-0395, Japan; tikesurg@surg2.med.kyushu-u.ac.jp (T.I.); ysoejima@surg2.med.kyushu-u.ac.jp (Y.S.); 12Department of Hepatobiliary and Pancreatic Surgery Shulan Hospital, Shulan Health Zhejiang University Hospital, Hangzhou 310014, China; yangzhe_0201730@163.com (Y.Z.); shusenzheng@zju.edu.cn (S.-S.Z.)

**Keywords:** disease progression, loco-regional therapy, alpha-fetoprotein, mRECIST, MELD, Metroticket 2.0

## Abstract

Since the introduction of Milan Criteria, all scoring models describing the prognosis of hepatocellular cancer (HCC) after liver transplantation (LT) have been exclusively based on characteristics available at surgery, therefore neglecting the intention-to-treat principles. This study aimed at developing an intention-to-treat model through a competing-risk analysis. Using data available at first referral, an upper limit of tumor burden for downstaging was identified beyond which successful LT becomes an unrealistic goal. Twelve centers in Europe, United States, and Asia (Brussels, Sapienza Rome, Padua, Columbia University New York, Innsbruck, Medanta-The Medicity Dehli, Hong Kong, Kyoto, Kaohsiung Taiwan, Mainz, Fukuoka, Shulan Hospital Hangzhou) created a Derivation (n = 2318) and a Validation Set (n = 773) of HCC patients listed for LT between January 2000–March 2017. In the Derivation Set, the competing-risk analysis identified two independent covariables predicting post-transplant HCC-related death: combined HCC number and diameter (SHR = 1.15; *p* < 0.001) and alpha-fetoprotein (AFP) (SHR = 1.80; *p* < 0.001). WE-DS Model showed good diagnostic performances at internal and external validation. The identified upper limit of tumor burden for downstaging was AFP ≤ 20 ng/mL and up-to-twelve as sum of HCC number and diameter; AFP = 21–200 and up-to-ten; AFP = 201–500 and up-to-seven; AFP = 501–1000 and up-to-five. The WE-DS Model proposed here, based on morphologic and biologic data obtained at first referral in a large international cohort of HCC patients listed for LT, allowed identifying an upper limit of tumor burden for downstaging beyond which successful LT, following downstaging, results in a futile transplantation.

## 1. Introduction

For a quarter of a century, morphologic Milan Criteria (MC) have been the cornerstone to select patients with hepatocellular cancer (HCC) for liver transplantation (LT) [1]. Many Western and Eastern centers successfully overruled these stringent criteria and were able to transplant a higher number of patients without increasing the risks of post-LT recurrence and death [2,3,4,5,6]. More recently, several variables looking at tumor biology have been introduced into clinical practice [7,8,9,10]. The combination of dynamic biological and morphological tumor characteristics following different downstaging (DS) therapies emerged as a promising tool to fine-tune the selection process of LT candidates in terms of delisting and post-LT recurrence [11,12,13,14,15,16,17,18]. In this context, the identification of an upper limit of tumor burden for DS, evaluated at first referral, is still a matter of debate. Apart from limited experiences, most prognostic HCC-LT models are based on criteria available at LT, therefore failing to consider the intention-to-treat (ITT) principle, namely failing to capture the risk of death or dropout during the waiting time period [11,14,15,16,17]. Only a few groups aimed at identifying well-defined eligibility criteria for DS; among them, the UNOS-downstaging (UNOS-DS) criteria are the most commonly adopted [19,20]. Conversely, many centers apply a so-called “all-comers” policy, enrolling potential HCC-recipients in a DS protocol without defining an upper limit of tumor burden [21]. Consequently, a better identification of a clinically applicable upper limit of DS should be of great importance with the intent to minimize the potential risk of futile LT, namely, a too high risk of tumor recurrence after transplant.

Moreover, the possibility to identify patients at high risk for futile transplant or, on the contrary, requiring a fast-track approach should be of great relevance in the donor selection process and in the setting of living donation program expansion [22].

This study aims to develop an ITT model enabling to predict the risk of dying after LT due to HCC recurrence based on variables available at first referral and also to identify an upper limit of tumor burden. We hypothesize that a successful LT after DS should become an unrealistic goal in patients exceeding this limit. The model was constructed using derivation and validation cohorts from centers belonging to an Eastern-Western collaborative HCC-LT consortium.

## 2. Results

The clinical and tumor characteristics of the entire population are displayed in Table 1. At first referral, 965 of 3091 (31.2%) patients did not meet MC. A total of 2369 (76.6%) patients were treated with at least one LRT before delisting or LT, and 1014 (32.8%) patients had LDLT. Hepatitis C virus (HCV)-related cirrhosis was the most common indication for listing (n = 1396; 45.2%). The only statistically significant difference found between derivation and validation data sets was the waiting time (3.7 vs. 4.5 months, respectively; *p* = 0.008). During a median follow-up of 40.8 months (IQR: 18.0–80.0), 2735 (88.5%) patients were transplanted, 303 (9.8%) were delisted, and 53 (1.7%) were still on the waiting list at the end of the study period. Delisting due to tumor progression or post-LRT liver failure occurred in 177 (5.7%) patients. HCC-independent delisting due to liver failure or death while on the waiting list occurred in 73 (2.4%) and 53 (1.7%) patients, respectively. Overall, 367 (13.4%) of 2735 transplanted patients experienced HCC recurrence, after a median time of 13.0 months (IQR: 7.0–28.5); 246 (9.0%) died, and 121 (4.4%) were still alive at last follow-up. Dead for other causes occurred in 452 (16.5%) patients.

Cumulative incidences of being delisted, being transplanted, dying of HCC recurrence, and dying after LT due to other causes are depicted in Figure 1.

### 2.1. Development of the WE-DS Model

The results of the multivariable competing-risk regression built using the Derivation Set data (n = 2318), are displayed in Table 2. Ten variables available at first entry were investigated: the number of lesions, the diameter of the largest lesion, AFP, Western center localization, MELD, previous LRT, HBV status, age, living-donor LT, HCV status and period of first referral (before vs. after 2010). Only variables obtained at first referral were used for constructing the WE-DS Model. This model identified two independent risk factors for post-transplant HCC-related death: number of lesions plus diameter of the largest lesion (SHR = 1.15; *p* < 0.001) and AFP level (SHR = 1.87; *p* < 0.001).

Graphical representations of the Model are visualized in Figure 2 as contour plots. The WE-DS Model (first referral time point; Figure 2A) was compared with the Metroticket 2.0 Score (LT time point; Figure 2B).

### 2.2. Internal and External Validation of the WE-DS Model

Both the Derivation (n = 2318) and Validation (n = 773) Sets were used for the evaluation of the diagnostic ability of the WE-DS Model in terms of risk of post-transplant HCC-related death (Table 3).

At internal validation, the WE-DS Model exhibited the highest c-statistic (AUC = 0.70; 95% CI = 0.67–0.74; *p* < 0.001). Metroticket 2.0, HALTHCC Score, Pre-MORAL Score, AFP-French Model, and MC status showed an AUC ranging 0.63–0.60; all the other tested scores had an AUCs <0.60.

The external validation confirmed the good diagnostic ability of the WE-DS Model (AUC = 0.66; 95% CI = 0.58–0.74; *p* < 0.001). Metroticket 2.0, HALTHCC Score, and AFP-French Model showed a slightly inferior AUC of 0.63; again, all other scores had an AUCs <0.60.

### 2.3. Identification of the Upper Tumor Burden Limit for Downstaging

The relationship between the Metroticket 2.0 calculated at LT time and the WE-DS Model evaluated at first referral is plotted in Figure 3A. Mazzaferro et al. set the previously reported upper limit of acceptable survival calculated from the time of LT at ≤30% [18]. After recalibration from the time of first referral, this acceptable risk was set to ≤13%. According to these results, an upper limit of tumor burden was identified (Figure 3B) based on four incremental combinations of morphology and AFP values. In particular, a patient would be considered meeting the WE-DS Model (and thus evaluable for potential transplantation) when meeting the following combinations: AFP ≤20 ng/mL and up-to-twelve as the sum of diameter and number of tumor lesions; AFP = 21–200 ng/mL and up-to-ten; AFP = 201–500 ng/mL and up-to-seven; AFP = 501–1,000 ng/mL and up-to-five.

When comparing the WE-DS Model to the MC and UNOS-DS criteria for the entire (Derivation and Validation Set) population of 3091 listed patients, no significant differences were observed in terms of dropout rates (8.5% to 10.0%) with the only exclusion of the WE-DS-OUT cases (12.1%; log-rank *p* = 0.003). After removing the cohort of 1533 Asian HCC-patients, in which no dropouts were reported, the differences among the Western centers raised, with 30.4% of dropouts in the WE-DS-OUT vs. 13.0% to 16.5% observed using the other criteria (log-rank *p* = 0.001).

Substantial differences were also reported when the 2735 transplanted patients were analyzed. The 5-year post-LT HCC-related death rate was 9.0% in the MC-IN, 15.0% in the MC-OUT/UNOS-DS-IN, 23.7% in the “all-comers” UNOS-DS-OUT, 14.4% in the MC-OUT/WE-DS-IN, and 37.7% in the “all-comers” WE-DS-OUT group, respectively (Figure 4). Interestingly, although UNOS-DS and WE-DS Model showed similar survival rates (log-rank *p* = 0.71), the latter model improved the selection and allowed more patients being transplanted. LT was performed in 669/2,735 (24.5%) initially MC-OUT/WE-DS-IN patients vs. 459 (16.8%) MC-OUT/UNOS-DS-IN cases (*p* < 0.001). Moreover, if the 5-year 30% survival cut-off for HCC-related death proposed by Mazzaferro was considered (dashed line in Figure 4), all groups presented results within this threshold except for the WE-DS-OUT group, confirming thereby the better selection ability of the here proposed model.

## 3. Discussion

The primary objective of the study was to predict in a “weighty manner” the risk of post-LT HCC-related death in transplant candidates, with the intent to identify an upper limit of DS and to standardize the LT inscription approach based on a large HCC-LT database including collaborating expert Western and Eastern centers. The currently available scoring systems mainly focus on the estimation of the risk of post-transplant HCC-related death using variables known at the time of LT. Therefore, information on intention-to-treat survival is lacking [1,2,3,4,5,6,11,12,13,14,15,16,18]. To avoid this shortcoming, data available at the first referral time-point were investigated. First referral was judged to be a more useful, reliable and reflecting-the-real period for prognostication of survival. In fact, the preliminary decisions to include a patient in a LT project, or to treat him with LRT, are taken at the time of the first encounter in the out-patient clinic.

The WE-DS Model presented here showed a high ability to select patients with a reduced chance to be successfully transplanted after DS therapies. DS has recently been shown to be a useful prognosticator as well as an identifier of patients presenting favorable tumor biology. Several studies demonstrated that successful DS allows obtaining similar post-LT survival rates respect to conventional LT criteria [8,9,10,19,20]. However, controversies still exist concerning the optimal upper limit of tumor burden to use for DS. According to Mazzaferro’s paper aiming at “squaring the circle” of HCC selection and allocation for LT, effectively downstaged patients should obtain a higher priority for LT, thereby minimizing the risk of dropout during the waiting time. However, the initial acceptance of “all-comers” needs to be linked with the potentially unacceptable risk of transplant futility (i.e., very high post-transplant HCC-related death rates) [23]. Data about the acceptable upper limit of tumor burden for entering in a liver transplant program are very scarce [19,20]. With the intent to standardize criteria for downstaging in the United States, UNOS recently implemented the UCSF/Region 5 DS-protocol as a new national policy for granting MELD exception for LT [24].

A recently published study investigating this UNOS-DS protocol showed that the “all-comers” presented a significantly lower rate of successful DS, a higher probability of dropout from the waiting list, and a lower 5-year ITT survival when compared to patients meeting the UCSF-DS criteria [25].

Another US study, including 3819 patients, identified in a multivariable analysis a short-to-mid waiting time (HR = 3.1; *p* = 0.005) and an AFP ≥100 ng/mL at LT (HR = 2.4; *p* = 0.009) as risk factors for post-transplant death [21]. According to these results, additional refinements based on AFP and waiting time were suggested in order to optimize the discriminatory ability of the UNOS-DS score further [21].

The WE-DS Model represents the first attempt to create a score integrating tumor morphology and biology to select the upper limit of DS. The advantage of integrating tumor morphology and biology is based on the concept that patients with high tumor number, diameter, or AFP at baseline are less likely to be able to be down-staged, or have tumor progression after initial down-staging and thus ultimately dropout or experience a high recurrence rate after LT [21]. An advanced statistical analysis, based on three different competing events, allowed constructing a mathematically robust model able to simultaneously report the estimations of receiving a transplant and of dying due to HCC recurrence, a piece of information sharply differing from the unique evaluation of dying from tumor-related causes after LT [18]. Moreover, the large Western-Eastern sample size and the relevant number of events (namely, post-LT HCC-related deaths) further allowed doing a robust statistical analysis.

The variables composing the WE-DS Model, namely HCC number, diameter, and AFP, were all identical to those observed in previously reported scores [11,12,13,18]. All of them are well-recognized risk factors for post-transplant HCC-related death and, most importantly, they all are available at patient first referral.

The inclusion of HCC patients belonging to the Western and Eastern world strengthens in our opinion the observed results, representing a unique approach to confirm the universal usability of the proposed scoring system. In fact, locally developed scores might be unbalanced by regional characteristics.

The reported results further reveal that patients out of the WE-DS Model had a significantly higher dropout rate, particularly in the Western cohort. This observation can be explained by the low incidence of LDLT in this geographical area, leading to longer waiting times and, therefore, higher dropout rates. Importantly, when only transplanted cases were analyzed, “all-comers” beyond the WE-DS Model was the only cohort to have five-year HCC-related death rates surpassing 30%, a value corresponding to a futile LT [18]. Interestingly, “all-comers” out of UNOS-DS remained below this 30% threshold, indicating a suboptimal stratification performance.

When the diagnostic ability for the risk of HCC-related post-LT death was internally and externally tested, the WE-DS Model had the best discriminatory power when compared to several previously proposed “urgency” or “utility” scores, all of them based on data obtained at the time of LT [1,2,3,4,5,6,11,12,13,14,15,16,18,19]. The Metroticket 2.0, the AFP French Model and the HALTHCC also had an excellent diagnostic performance, mainly revealed in the Validation Set [11,12,18]. Interestingly, all of these three scores are composed by the same three variables, namely AFP, tumor number, and diameter.

Conversely, as expected, the “urgency” models UNOS-DS, DeMELD, MELD-EQ and HCC-MELD performed worst, because they were constructed with the intent to identify dropout rates rather than post-transplant HCC-related deaths [14,15,16].

This study can be criticized by comparing the WE-DS Model, based on data obtained at first referral, to scores based on data available at time of transplantation. Unfortunately, no other study investigated the prognostic strength of a model exclusively based on ‘first come’ variables. Only the French AFP Model included data obtained at the time of listing [11]. The TRAIN score, also based on intention-to-treat data [17], was not tested here because based on “dynamic” AFP and tumor radiological modifications following LRT (and thus not at first referral).

When the WE-DS Model was compared with the currently adopted MC and UNOS-DS criteria [1,19] in terms of dropout and post-LT HCC-related death, it was interesting to observe that only the WE-DS-OUT patients showed poor results. In detail, if the 5-year 30% survival cut-off for HCC-related death proposed by Mazzaferro was considered [18], only the WE-DS-OUT cases exceeded this value, further underlying the selection ability of the WE-DS model.

The present study has some limitations. Firstly, the heterogeneity of the investigated population may be addressed as the main drawback of the study, due to the different match/allocation systems and LRT approaches used among the various centers. In fact, we feel that these differences may represent a benefit concerning the design of the study, namely the creation of a mathematical model based only on internationally relevant risk factors, avoiding thereby center-related biases. Secondly, the retrospective nature of the study potentially affects the power to appraise variables such as radiological characteristics. Such limitation is shared with all the studies including large HCC patient cohorts [4,10,11,12,14,15,16,17,18]. The considerable variability of treatment management among centers might represent a third point of weakness. As this study relies on first-referral data only, differences in LRT strategies are bypassed; moreover, the use of LRT therapies did not affect any of the competing-risk models. This may be explained by the fact that post-LRT response rather than LRTs per se influences delisting and post-transplant recurrence risk [17]. Unfortunately, the decision to use only first-referral variables for the construction of the model limited our opportunity to capture the prognostic role of response to bridging or downstaging LRT. Several studies revealed that radiological response to LRT is a better predictor of tumor biology than tumor morphology alone [15,17,25]. Although it is understandable that constructing a model without radiological response to LRT is a limitation, it was voluntarily decided not to investigate this parameter. In fact, radiological response after LRT is available only at time of LT, making impossible to use this variable for constructing a model based only on “first referral” variables.

## 4. Materials and Methods

### 4.1. Patients

A retrospective analysis of the data from 3325 adult (≥18 years) patients with a diagnosis of HCC at the time of first referral for LT and referred during the period 1 January 2000–31 March 2017 was performed. In detail, the participating centers with the corresponding number of patients enrolled in the present study were: Padua (Italy; n = 630), New York Columbia University/Weill Cornell Medical Center (USA; n = 356), Brussels (Belgium; n = 332), Hong Kong (SAR of People’s Republic of China; n = 315), Innsbruck (Austria; n = 313), New Delhi (India; n = 270), Rome Sapienza University (Italy; n = 248), Kyoto (Japan; n = 230), Taiwan (Republic of China; n = 200), Mainz (Germany; n = 176), Kyushu (Japan; n = 161), and Hangzhou Shulan Health Hospital/First Affiliated Hospital (People’s Republic of China; n = 94).

Preliminary study protocol, data collection management, and study coordination among the involved centers were realized by JL (Scientific Coordinator, Brussels) and QL (Data Manager, Sapienza Rome).

### 4.2. Study Design

All the patients consecutively evaluated and enlisted for LT with a radiological diagnosis of HCC were considered recruitable for the present study. Nature of LT (upfront vs. salvage) or type and number of neo-adjuvant treatments were not considered exclusion criteria. After the exclusion of patients with mixed hepatocellular-cholangiocellular cancer, cholangiocarcinoma misdiagnosed as HCC, and incidental HCC, the studied sample numbered 3091 cases. A Derivation Set of 2318 candidates (75.0%) and a Validation Set of 773 candidates (25.0%) were obtained from the entire population. Block randomization was performed to maintain a similar representation of Western and Eastern centers in the two data sets (Appendix A).

### 4.3. Hepatocellular Cancer Management and Definitions

The diagnosis of HCC was made according to international guidelines [26,27,28]. Tumor upper limits for transplantability significantly differed among the centers, with a more conservative approach in the West and a more aggressive approach in the East, namely with higher percentages of enlisted patients initially out of the conventional LT criteria (Appendix A).

The decision to perform loco-regional treatments (LRT) was also different among the centers, although the possibility of down-staging or bridging was shared in case of expected long waiting times [26,27,28]. Several variables were collected at the time of first referral and of LT or delisting with the intent to construct a comprehensive ITT model using only first-referral variables. The diameter of the largest tumor and the number of nodules were evaluated taking only into account the vital tissue as identified by arterial enhancement. Response to neoadjuvant therapy was prospectively assessed using the modified Response Evaluation Criteria in Solid Tumours (mRECIST) in patients listed after 2010. In patients enlisted before 2010, mRECIST were evaluated retrospectively by local radiologists.

The Institutional Ethical and Scientific Review board of the coordinating center approved the study (Sapienza University of Rome, approval code 1214/2019). All the other centers obtained local approvals according to the ethical rules concerning retrospective studies. The study was registered at http://www.ClinicalTrials.gov (ID: NCT03595345). All data in relation to liver transplantations performed in the People’s Republic of China were obtained after the modification of the transplant law banning the use of organs from executed prisoners (year 2015).

### 4.4. Statistical Analysis

Continuous variables were reported as medians and interquartile ranges (IQR). Dummy variables were reported as numbers and percentages. The maximum likelihood estimation method for managing missing data was used [29]. For all the variables used for constructing the models, the missing data were always <5%. Mann–Whitney U test and Fisher’s exact test were used for comparisons of continuous and categorical variables, respectively.

The model was constructed using the Derivation Set data and adopting the Fine and Gray methodology for competing-risk regressions [30]. Sub-hazard ratios (SHR), and 95% confidence intervals (95%CI) were reported. Three competing events were investigated: (1) delisting; (2) death after LT due to HCC recurrence, and (3) death after LT due to no-HCC-related causes. The delisting was defined as any event of dropout or death whilst on the waiting list. Post-transplant HCC-related death was defined as any death directly related to tumor recurrence after LT. All patients who died of causes other than HCC or were alive with recurrence at the date of the last follow-up visit were censored. The last censoring was performed on 31 March 2017. Our attention was focused on the second model, namely, the risk of dying after LT due to tumor recurrence. The accuracy of the model was assessed in the Derivation and Validation Set through c-statistics. Confidence intervals for the c-statistic derived from 100 bootstrap replications of the technique. The accuracy of the model was compared with several previously proposed criteria able to predict the risk of dropout or post-transplant recurrence [1,2,3,4,5,6,11,12,13,14,15,16,17,18].

After adopting the results obtained from the proposed model, an upper limit of tumor burden for DS was identified. The five-year acceptable risk for post-LT HCC-related death ≤30% proposed in the Metroticket 2.0 study was considered [18]. This value was “recalibrated” in an ITT fashion. This “recalibration” allowed identifying a tumor burden threshold based on a combination of alpha-fetoprotein (AFP) level, tumor number, and diameter, all of them being available at first referral. These parameters participated in creating the West–East DownStaging Model (WE-DS). A more detailed explanation of this estimation is reported in Appendix A.

The Kaplan–Meier method was used with the intent to evaluate the observed dropouts and post-transplant HCC-related deaths. Different groups determined according to different tumor burden available at first referral were compared: (1) MC-IN; (2) MC-OUT/UNOS-DS-IN; (3) all-comers out of UNOS-DS; (4) MC-OUT/WE-DS-IN; (5) all-comers out of WE-DS.

Variables with a *p* < 0.05 were considered statistically significant. SPSS statistical package version 24.0 (SPSS Inc., Chicago, IL, USA) was used and competing-risk analyses were done using the packages “cmprsk”, “risk-regression”, “crrSC”, and “pec” of R-project (R version 3.4.3; R Foundation for Statistical Computing, Vienna, Austria).

## 5. Conclusions

In conclusion, the WE-DS Model, based on both morphologic and biologic data obtained at first referral in a large international (Western-Eastern) cohort of HCC patients listed for LT, allowed identifying an upper limit of tumor burden for downstaging beyond which successful LT, following downstaging, results in a futile transplantation.

## Figures and Tables

**Figure 1 cancers-12-00452-f001:**
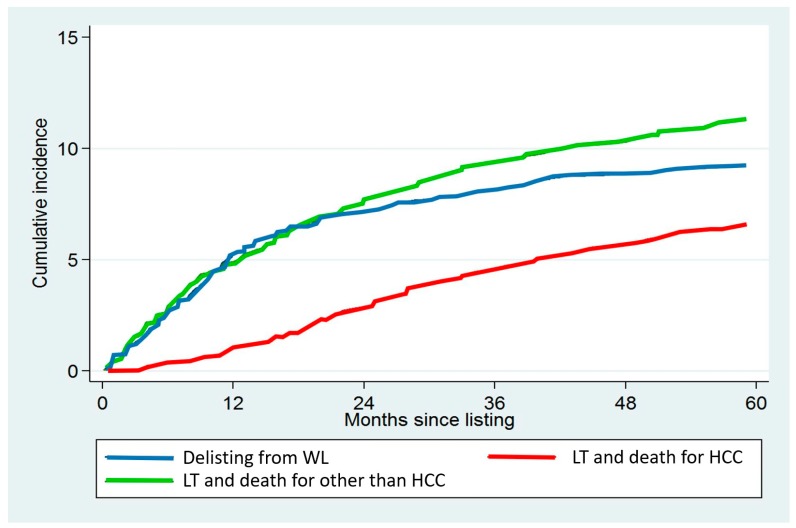
Cumulative incidences of delisting, of receiving LT and dying for HCC, and of receiving LT and dying for other causes, in the entire study population of 3091 patients.

**Figure 2 cancers-12-00452-f002:**
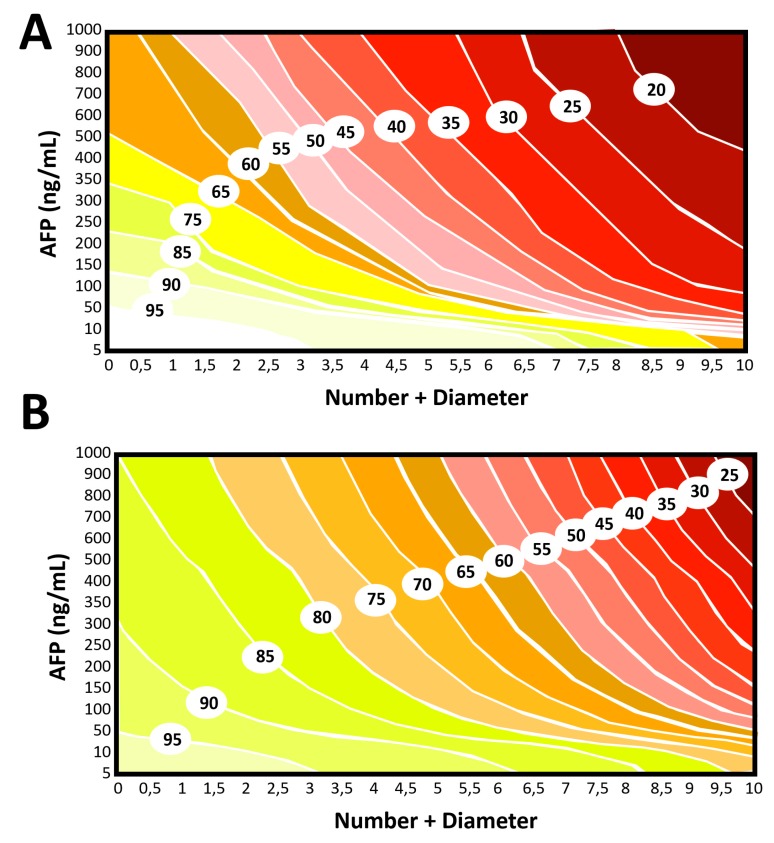
(**A**) Contour plot showing the five-year risk of post-transplant HCC-related survival estimated using the present “first referral” model. (**B**) Contour plot showing the five-year risk of post-transplant HCC-related survival estimated using the Metroticket 2.0 Score (variables evaluated at the time of LT).

**Figure 3 cancers-12-00452-f003:**
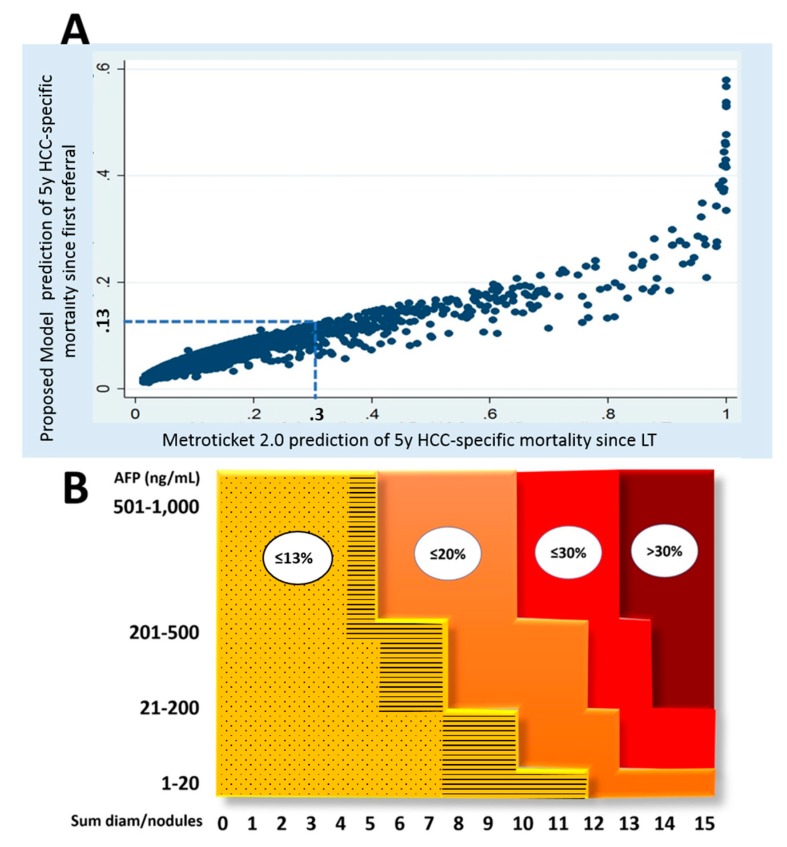
(**A**) Relationship between the original predictions of Metroticket 2.0 at LT (x-axis) and those recalculated in the present model since first referral (y-axis). The cumulative five-year risk of post-transplant HCC-related death estimated with the Metroticket 2.0 at LT = 30% corresponded to a risk calculated with the model since listing = 13% (dashed lines). (**B**) Simplified model showing three different thresholds according to the post-transplant HCC-specific death risk = 13%, 20%, and 30%. The yellow area with dashed points corresponded to the “transplantable” area according to the Metroticket 2.0, while the yellow area with lines corresponded to the downstaging area according to the present model.

**Figure 4 cancers-12-00452-f004:**
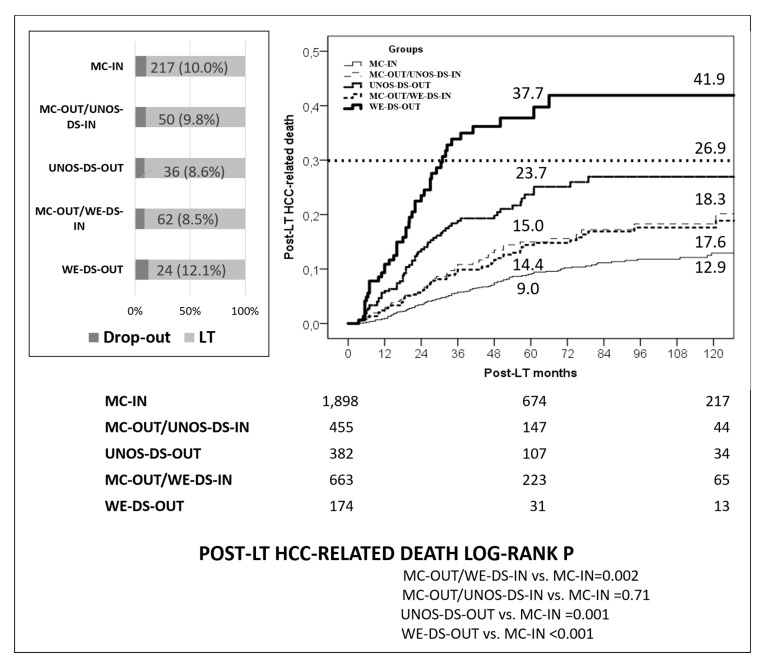
Dropout and post-LT HCC-related death rates in the MC-IN, MC-OUT/UNOS-DS-IN, UNOS-DS-OUT, MC-OUT/WE-DS-IN, and WE-DS-OUT patients. Dashed line identifies the post-LT 5-year 30% of HCC-related death proposed by Mazzaferro [18].

**Table 1 cancers-12-00452-t001:** Demographics and tumor characteristics in the Derivation and Validation Sets.

Variables	Entire Population	Derivation Set	Validation Set	*p*-Value
	(N = 3091; 100.0%)	(N = 2318; 75.0%)	(N = 773; 25.0%)	
	Median (IQR) or n (%)			
**Age at first referral (years)**	**58 (53–63)**	**58 (53–63)**	58 (53–63)	0.5
Male gender	2477 (80.1)	1864 (80.4)	613 (79.3)	0.5
MELD score at first referral	12 (9–16)	12 (9–16)	12 (9–16)	0.2
Living donation	1014 (32.8)	753 (32.5)	261 (33.8)	0.5
West/East	1912/1179 (61.9/38.1)	1437/881 (62.0/38.0)	475/298 (61.4/38.6)	0.8
Cause of cirrhosis (*)				
Hepatitis C	1396 (45.2)	1046 (45.1)	350 (45.3)	1.0
Hepatitis B	890 (28.8)	674 (29.1)	216 (27.9)	0.6
Alcohol	637 (20.6)	467 (20.1)	170 (22.0)	0.3
NASH-cryptogenic	208 (6.7)	161 (6.9)	47 (6.1)	0.5
Other	146 (4.7)	109 (4.7)	37 (4.8)	0.9
Waiting time from first referral (months)	4.2 (1.5–10.0)	4.5 (1.5–10.5)	3.7 (1.5–9.0)	0.008
HCC radiological features at first referral				
Target lesion (cm)	2.5 (1.8–3.8)	2.5 (1.8–3.8)	2.5 (1.7–3.9)	0.6
Number of nodules	1 (1–3)	1 (1–3)	1 (1–3)	0.7
MC-OUT status	965 (31.2)	720 (31.1)	245 (31.7)	0.8
LRT before LT/delisting (*)	2369 (76.6)	1783 (76.9)	586 (75.8)	0.5
Number of treatments	2 (1–3)	2 (1–3)	2 (1–3)	0.6
TACE	1838 (59.5)	1377 (59.4)	461 (59.6)	0.9
RFA/PEI	1228 (39.7)	933 (40.3)	295 (38.2)	0.3
TARE/EBRT/ HIFU/micro-waves	300 (9.7)	231 (10.0)	69 (8.9)	0.4
Hepatic resection	269 (8.7)	205 (8.8)	64 (8.3)	0.7
Last HCC radiological features (vital tissue)				
Target lesion (cm)	2.2 (1.3–3.3)	2.2 (1.4–3.3)	2.2 (1.3–3.3)	0.8
Number of nodules	2 (1–3)	2 (1–3)	1 (1–3)	0.5
MC-OUT status	920 (29.8)	680 (29.3)	240 (31.0)	0.4
Radiologic response at last assessment				
Complete response	277 (9.0)	213 (9.2)	64 (8.3)	0.5
Partial response	976 (31.6)	731 (31.5)	245 (31.7)	0.9
Stable disease	384 (12.4)	290 (12.5)	94 (12.2)	0.9
Disease progression	732 (23.7)	549 (23.7)	183 (23.7)	1.0
No LRT	722 (23.4)	535 (23.1)	187 (24.2)	0.5
AFP at HCC diagnosis (ng/mL)	12.6 (5.0–50.0)	12.0 (5.0–50.0)	14.5 (5.0–52.5)	0.3
AFP at last assessment (ng/mL)	10.7 (4.1–60.0)	10.4 (4.1–61.6)	11.6 (4.4–55.0)	0.6
Delisting	303 (9.8)	224 (9.7)	79 (10.2)	0.7
HCC-related dropout	177 (5.7)	128 (5.7)	49 (6.5)	0.4

Derivation (n = 2318; 75.0%) and Validation Set (n = 773; 25.0%) were obtained from the entire population. Block randomization was performed to maintain a similar representation of Western and Eastern centers in the two data sets. (*) In some cases, combined etiology of the underlying cirrhosis or multiple LRT approaches in the same patient. Abbreviations: IQR, interquartile ranges; n, number; LT, liver transplantation; MELD, model for end-stage liver disease; NASH, non-alcoholic steatohepatitis; HCC, hepatocellular cancer; MC, Milan Criteria; LRT, loco-regional treatments; TACE, trans-arterial chemo-embolization; RFA, radio-frequency ablation; PEI, percutaneous ethanol injection; TARE, trans-arterial radio-embolization; EBRT, external beam radiotherapy; HIFU, high intensity focused ultrasounds; AFP, alpha-fetoprotein.

**Table 2 cancers-12-00452-t002:** Model for the risk of post-transplant HCC-related death.

Variable	Beta	SHR	95%CI		*p*	Beta	SHR	95%CI		*p*
			Lower	Upper				Lower	Upper	
	Univariable Model					Multivariable Model				
**Number + dimension at entry**	**0.16**	**1.18**	**1.13**	**1.23**	**<0.001**	**0.14**	**1.15**	1.11	1.20	<0.001
AFP at entry	0.66	1.94	1.69	2.23	<0.001	0.63	1.87	1.62	2.16	<0.001
West provenience (vs. East)	0.03	1.03	0.77	1.38	0.84	-0.32	0.73	0.51	1.03	0.08
MELD at entry (per unit)	−0.02	0.98	0.95	1.01	0.19	−0.02	0.98	0.95	1.01	0.21
LRT	0.03	1.03	0.73	1.45	0.86	−0.005	0.995	0.69	1.44	0.98
HBV	0.02	1.02	0.74	1.39	0.92	0.13	1.14	0.82	1.58	0.45
Age at entry (per year)	−0.02	0.98	0.96	1.00	0.04	−0.006	0.99	0.98	1.01	0.52
LDLT	0.10	1.11	0.82	1.49	0.51	−0.03	0.98	0.72	1.31	0.87
HCV	0.01	1.01	0.76	1.34	0.97	0.03	1.03	0.75	1.40	0.87
Period of fist referral ≤2010	0.03	1.03	0.75	1.41	0.85	0.0001	1.00	0.71	1.41	0.998

SHR, sub-hazard ratios; CI, confidence intervals; AFP, alpha-fetoprotein; MELD, model for end-stage liver disease; LDLT, living-donor liver transplantation; HBV, hepatitis B virus; HCV, hepatitis C virus; LRT, loco-regional therapy.

**Table 3 cancers-12-00452-t003:** Comparison between different scores for the prediction of post-transplant HCC-related death, in the Derivation (n = 2318) and Validation Cohort (n = 773).

Variable	Derivation Cohort (n = 2318)					Validation Cohort (n = 773)				
	AUC	SE	95%CI		*p*	AUC	SE	95%CI		*p*
			Lower	Upper				Lower	Upper	
WE-DS ≤13%	0.70	0.02	0.67	0.74	<0.001	0.66	0.04	0.58	0.74	<0.001
Metroticket 2.0 ≤30%	0.62	0.02	0.57	0.66	<0.001	0.63	0.04	0.55	0.71	0.001
HALTHCC ≤17	0.62	0.02	0.58	0.67	<0.001	0.63	0.04	0.55	0.71	0.001
Pre-MORAL ≤6	0.62	0.02	0.58	0.67	<0.001	0.59	0.04	0.53	0.66	0.009
AFP-French model ≤2	0.61	0.02	0.57	0.66	<0.001	0.63	0.04	0.55	0.72	0.001
MC-IN	0.60	0.02	0.56	0.65	<0.001	0.54	0.04	0.46	0.62	0.28
Up-to-seven ≤7	0.58	0.02	0.53	0.62	<0.001	0.56	0.04	0.48	0.64	0.14
Asan-IN	0.57	0.02	0.53	0.62	0.001	0.59	0.04	0.50	0.67	0.03
UNOS-DS-IN	0.57	0.02	0.52	0.61	0.002	0.56	0.04	0.48	0.65	0.11
Kyoto-IN	0.57	0.02	0.52	0.61	0.002	0.58	0.04	0.49	0.66	0.05
UCSF-IN	0.57	0.02	0.53	0.62	0.001	0.55	0.04	0.47	0.63	0.20
deMELD	0.55	0.02	0.51	0.60	0.02	0.50	0.04	0.42	0.57	0.91
Hangzhou-IN	0.53	0.02	0.48	0.57	0.19	0.53	0.04	0.45	0.61	0.45
MELDEQ ≤15	0.52	0.02	0.47	0.56	0.42	0.49	0.04	0.42	0.57	0.88
HCC-MELD	0.44	0.02	0.40	0.49	0.01	0.41	0.04	0.33	0.49	0.02

*p* values estimated against the fixed value of 0.50. AUC, area under the curve; SE, standard error; CI, confidence intervals; WE-DS, West–East Downstaging; HALTHCC, Hazard Associated with Liver Transplantation for Hepatocellular Carcinoma; MORAl, Model of recurrence after liver transplant; AFP, alpha-fetoprotein; MC, Milan Criteria; UNOS-DS, United Network for Organ Sharing; UCSF, University of California San Francisco; deMELD, dropout equivalent Model for End-stage Liver Disease; MELDEQ, Model for End-stage Liver Disease equivalent; HCC-MELD, hepatocellular carcinoma - Model for End-stage Liver Disease.

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
