# Peer review of "Identification of an Upper Limit of Tumor Burden for Downstaging in Candidates with Hepatocellular Cancer Waiting for Liver Transplantation: A West–East Collaborative Effort"

_cancers, 2020, doi:10.3390/cancers12020452_

Round 1
Reviewer 1 Report
This is a multicenter retrospective study trying to address the ITT analysis for HCC liver transplant candidates at the moment of the first referral.
Overall the manuscript is well written and clear in the intention, although some aspects could be improved.
First, I would suggest to expand the introduction with regard to the effect of donor selection in the HCC subgroup.
Second, what was the criteria for selecting the participating centers and for DS in each of the participating center?
Third, the authors confirm that DS is a useful prognosticator for HCC recurrence, but did they compare the effect of different DS techniques, such as RFA vs TACE beyond RECIST? Was there any information regarding the actual necrosis achieved and confirmed on the explanted liver? Was the time to DS considered?
Fourth, biological and demographic differences between East and West recipients are huge; in conclusion the authors confirmed that AFP, HCC number and diameter relate to tumour recurrence. Is there any additional information recipient-derived that could be investigated in the ITT?
Author Response
Reviewer 1.
This is a multicenter retrospective study trying to address the ITT analysis for HCC liver transplant candidates at the moment of the first referral. Overall the manuscript is well written and clear in the intention, although some aspects could be improved.
Response: We thank Reviewer#1 for the positive comments and for the opportunity to improve the quality of the study.
First, I would suggest to expand the introduction with regard to the effect of donor selection in the HCC subgroup.
Response: We than Reviewer#1 for the intriguing question. We added in Introduction the following sentence: “Moreover, the possibility to identify patients at high risk for futile transplant or, on the opposite, requiring a fast-track approach should be of great relevance in the donor selection process and in the setting of living donation program expansion [22].”
Second, what was the criteria for selecting the participating centers and for DS in each of the participating center?
Response: The centers involved represent a combination of Western centers composing the EurHeCaLT Study Group and Eastern centers, all of them being involved in building up an expertise in the field of transplant oncology. All the collaborative centers also had similar approaches during the pre-transplant wait list period, namely implementing an aggressive strategy of neo-adjuvant treatments increasing the repeated application of different locoregional treatments.
As reported in the text, no common criteria were present in different centers. Overall, the internationally recognized guidelines were followed by all the centers. We reported the following sentence: “The decision to perform loco-regional treatments (LRT) was also different among the centers, although the possibility of down-staging or bridging was shared in case of expected long waiting times [22-24].”
Third, the authors confirm that DS is a useful prognosticator for HCC recurrence, but did they compare the effect of different DS techniques, such as RFA vs TACE beyond RECIST? Was there any information regarding the actual necrosis achieved and confirmed on the explanted liver? Was the time to DS considered?
Response: We thank Reviewer#1 for this intriguing question. We have all of the data required, but we did not decide to investigate in detail all of these aspects because we considered them out of the specific intents of the study. Our idea in this particular study was to construct a model able to predict when we need to do a DS, instead of observing the results of the DS itself.
The proposal of Reviewer#1 to investigate all of the data concerning the different use of LRT and the necrosis results is extremely interesting.
We already investigated these aspect in the setting of bridging therapies in case of Milan-IN patients (Lai Q, Vitale A, Iesari S, Finkenstedt A, Mennini G, Onali S, Hoppe-Lotichius M, Manzia TM, Nicolini D, Avolio AW, Mrzljak A, Kocman B, Agnes S, Vivarelli M, Tisone G, Otto G, Tsochatzis E, Rossi M, Viveiros A, Ciccarelli O, Cillo U, Lerut J; European Hepatocellular Cancer Liver Transplant Study Group. The Intention-to-Treat Effect of Bridging Treatments in the Setting of Milan Criteria-In Patients Waiting for Liver Transplantation. Liver Transpl. 2019 Jul;25(7):1023-1033. doi: 10.1002/lt.25492).
We are confident to further investigate in this direction also in the specific setting of downstaging in the next studies.
Fourth, biological and demographic differences between East and West recipients are huge; in conclusion the authors confirmed that AFP, HCC number and diameter relate to tumour recurrence. Is there any additional information recipient-derived that could be investigated in the ITT?
Response: Apart AFP, HCC number and diameter, we investigated in the Fine-Gray Competing-Risk Model other eight variables, namely the place of transplant (West provenience vs. East), the MELD at entry, the previous use of LRT, HBV and HCV infection, age at entry, living donation, and first referral ≤2010. None of these variables was significant. It should be interesting to observe that all the “regional” variables (place of transplant, living donation, and specific viral infection) failed to be relevant. This datum should further justify the following sentence we placed in the Discussion part: “The present study has some limitations. Firstly, the heterogeneity of the investigated population may be addressed as the main drawback of the study, due to the different match/allocation systems and LRT approaches used among the various centers. We feel that these differences may represent a benefit concerning the design of the study, namely the creation of a mathematical model based only on internationally relevant risk factors, avoiding thereby center-related biases.”

Reviewer 2 Report
Title: Identification of an upper limit of tumor burden for downstaging in Journal: Cancers |
Thank you for inviting me to review the above titled manuscript. The paper looks interesting. However, there are several issues that need to be addressed by authors.
Abstract: The abstract could be strengthened. No need to state the clinical approval ID in the abstract. What are these 12 centres? State them. Conclusion should be rewritten. Introduction: Give an example, why the new model is needed. Clarify the intention-to-treat principles (lines 58/59). State a research question and a hypothesis. Material and methods: Line 70, state the place of data, country and city. The data date back to the year 2000, but at lines 344/346 the authors stated that data obtained after the modification of the transplant law banning the use of organs from executed prisoners. How can you explain this contradiction. Also, please state ethical approval from all 12 centres or representations from these centres. Lines 74/75: Why “incidental HCC” were excluded? Define validation set. State the study design before line 79 as a new sub-section (2.2) and change 2.2 to become 2.3 Line 80 {references 22, 23}- why the American Association for the Study of Liver Diseases were not used as well? Lines 82/83 {a more progressive approach in ….} explain conservative vs progressive approaches, give examples. Lines 87/88 – not clear with all differences mentioned, why this model should be based on the West-East despite all these differences? Methods is not complete. The authors should address 1) what are the data considered for inclusion? 2) How these data from East and West were searched and evaluated? Consistency issues, 3) How the coordination between centres was carried out? 4) What were the problems in the data collection? Lines 110-127, state the names of the 12 centres, and number or ratio of data collected from each. Method should state how the model was designed, how the model was validated, and how the upper tumour burden limit etc was identified/defined- you need to mirror issues in results with the methods. Results: Line 143- Table 1- define Derivation set and Validation set under the table. Table 1: The distribution between West and East re distribution of data not clear, need to be improved. Discussion: Lines 304/305: give a reference Line 304 “..with all the studies…” not correct. Why “all”. The language of the manuscript should be carefully reviewed. Lines 315/316: ‘In fact…” should be clarified. Not clear, Lines 345-347: ethical approval concerns, also we need ethical approval from other centres.
Author Response
Reviewer 2.
We thank Reviewer#2 for the positive comments and for the opportunity to improve the quality of the study.
Abstract:
The abstract could be strengthened.
Response: We thank Reviewer#2 for the comment. Unfortunately, the limited number of words for the abstract limited our possibility to clarify some aspects of the study better. We followed the recommendations of Reviewer#2 with the intent to strengthen the abstract.
No need to state the clinical approval ID in the abstract.
Response: We removed the approval ID accordingly.
What are these 12 centres? State them.
Response: We added the list of the involved centers.
Conclusion should be rewritten.
Response: We rewrote the conclusions trying to strengthen them.
Introduction:
Give an example, why the new model is needed.
Response: We thank Reviewer#2 for the pertinent comment. We added a sentence trying to clarify better why this new model is needed: “Consequently, better identification of a clinically applicable upper limit of DS should be of great importance with the intent to minimize the potential risk of futile LT, namely a too high risk of tumor recurrence after transplant.”
Clarify the intention-to-treat principles (lines 58/59).
Response: We thank Reviewer#2 for the comment. We added a sentence with the intent to clarify the concept of intention-to-treat in this specific setting better. We added the following sentence: “namely failing to capture the risk of death or drop-out during the waiting time period”.
State a research question and a hypothesis.
Response: We bedded clarified our question and hypothesis. We added the following sentence: “We hypothesize that a successful LT after DS should become an unrealistic goal in patients exceeding this limit.”
Material and methods:
Line 70, state the place of data, country and city.
Response: We clarified the numerosity of each center with specified country and city. We added the following sentence: “In detail, the participating centres with the corresponding number of patients enrolled in the present study were: Padua (Italy; n=630), New York Columbia University/Weill Cornell Medical Center (USA; n=356), Brussels (Belgium; n=332), Hong Kong (SAR of People’s Republic of China; n=315), Innsbruck (Austria; n=313), New Delhi (India; n=270), Rome Sapienza University (Italy; n=248), Kyoto (Japan; n=230), Taiwan (Republic of China; n=200), Mainz (Germany; n=176), Kyushu (Japan; n=161), and Hangzhou Shulan Health Hospital/First Affiliated Hospital (People’s Republic of China; n=94).”
The data date back to the year 2000, but at lines 344/346 the authors stated that data obtained after the modification of the transplant law banning the use of organs from executed prisoners. How can you explain this contradiction.
Response: The limitation involved the only center coming from the People’s Republic of China, namely the Shulan Health Zhejiang University Hospital of Hangzhou. In this case, we enrolled only the cases transplanted during the period 2015-2017. All the other centers reported the enlisted cases during the period 2000-17.
Also, please state ethical approval from all 12 centres or representations from these centres.
Response: This is a retrospective study. We obtained the ethical approval from the leader center, namely Sapienza University of Rome. All the other centers obtained local approvals according to the ethical rules concerning retrospective studies. No modification of diagnostic strategies, therapies nor any other clinical approach were done for the present study. The study was also registered on ClinicalTrials obtaining an ID code.
We added the following sentence: “The Institutional Ethical and Scientific Review board of the coordinating center approved the study (Sapienza University of Rome, approval code 1214/2019). All the other centers obtained local approvals according to the ethical rules concerning retrospective studies. The study was registered at http://www.ClinicalTrials.gov (ID: NCT03595345).”
Lines 74/75: Why “incidental HCC” were excluded?
Response: We removed the incidental HCCs because their diagnosis should be done only on explant pathology. The present study is based on the construction of a model constructed using radiological findings at first referral. As a consequence, introducing incidental HCC should represent a bias for the creation of the model, because these patients present no tumor at radiology.
Define validation set.
Response: We already explained how we obtained the Validation Set in Material and Methods in the following sentence. “A Derivation Set of 2,318 candidates (75.0%) and a Validation Set of 773 candidates (25.0%) were obtained from the entire population. Block randomization was performed to maintain a similar representation of Western and Eastern centers in the two data sets”. We better clarified how we did in the Supplementary Material. In detail, the entire population of 3,091 patients was split into a Derivation Set of 2,318 candidates (75.0%) and a Validation Set of 773 candidates (25.0%). A block randomization was performed to maintain a similar representation of Western and Eastern centres in the two data sets. The entire population was divided in twelve subgroups (“blocks”), each one corresponding to every centre participating in the study. Then, patients within each block were randomly assigned to the Derivation or the Validation Set using a casual number generator. We arbitrarily decided to use the 75.0% and the 25.0% of the entire population for constructing the two different sets.
State the study design before line 79 as a new sub-section (2.2) and change 2.2 to become 2.3
Response: We thank Reviewer#2 for the comment. We modified accordingly.
Line 80 {references 22, 23}- why the American Association for the Study of Liver Diseases were not used as well?
Response: We agree with Reviewer#2. We added the reference of the last AASLD guidelines.
Lines 82/83 {a more progressive approach in ….} explain conservative vs progressive approaches, give examples.
Response: We thank Reviewer#2 for the pertinent question. Indeed, the sentence is not clear. We preferred substituting the word “progressive” with the term “aggressive”. In other words, East centers enlist cases with more advanced cancers. We tried to better explain the concept modifying the sentence in the following way: “Tumor upper limits for transplantability significantly differed among the centers, with a more conservative approach in the West and a more aggressive approach in the East, namely with higher percentages of enlisted patients initially out of the conventional LT criteria”.
Lines 87/88 – not clear with all differences mentioned, why this model should be based on the West-East despite all these differences?
Response: We understand the concerns of Reviewer#2. We already reported this aspect as a limitation of the study. In detail, we stated that “The present study has some limitations. Firstly, the heterogeneity of the investigated population may be addressed as the main drawback of the study, due to the different match/allocation systems and LRT approaches used among the various centers.”
However, we also added that: “In fact, we feel that these differences may represent a benefit concerning the design of the study, namely the creation of a mathematical model based only on internationally relevant risk factors, avoiding thereby center-related biases.”
If we consider the geographical and management-related differences existing not only between East and West, but even in countries placed in the same Continent, the idea of realizing a unified model should appear hazardous. However, on the opposite, the possibility to perform an inferential analysis placing together the prerogatives of the different centers minimizes the impact of the “local” approaches, finally identifying the covariates effectively playing an important role in all the contexts. We hope the possibility to construct such a model should be of utility mainly due to the attempt to minimize the effect of the center-related biases.
Methods is not complete.
The authors should address 1) what are the data considered for inclusion?
Response: We added the following sentence: “All the patients consecutively evaluated and enlisted for LT with a radiological diagnosis of HCC were considered recruitable for the present study.”
2) How these data from East and West were searched and evaluated? Consistency issues,
Response: In the great majority of cases, we asked for data easy to be collected and internationally recognized in the setting of HCC. The main problem of consistency was related to the mRECIST evaluation, mainly in cases between 2000 and 2010. We asked for a local assessment by radiologists, with the intent to reclassify cases with RECIST, EASL, or other criteria for radiological response. We added the following sentence with the purpose of clarifying better this aspect: “Response to neoadjuvant therapy was prospectively assessed using the modified Response Evaluation Criteria in Solid Tumours (mRECIST) in patients listed after 2010. In patients enlisted before 2010, mRECIST were evaluated retrospectively by local radiologists”.
We cannot exclude a possible bias related to this process. However, we should underline the fact that our final model is not based on the radiological response. Consequently, we think that the impact of potential biases related to radiological response should be minimal or even nihil in this study. We reported this concept in discussion: ”Unfortunately, the decision to use only first-referral variables for the construction of the model limited our opportunity to capture the prognostic role of response to bridging or downstaging LRT. Several studies revealed that radiological response to LRT is a better predictor of tumor biology than tumor morphology alone [15,17,29]. Although it is understandable that constructing a model without a radiological response to LRT is a limitation, it was voluntarily decided not to investigate this parameter. In fact, response to LRT is merely obtained after the first referral, so a model constructed using this variable is therefore judged to be statistically inconsistent.”
3) How the coordination between centres was carried out?
Response: We thank Reviewer#2 for the question. With the intent to better clarify how the study coordination was done, we added the following sentence: “Preliminary study protocol, data collection management and study coordination among the involved centers were realized by JL (Scientific Coordinator, Brussels) and QL (Data Manager, Sapienza Rome).”
4) What were the problems in the data collection?
Response: We reported in the statistical part how we managed the problem of missing data. “The maximum likelihood estimation method for managing missing data was used [25]. For all the variables used for constructing the models, the missing data were always <5%.” We specifically avoided to use variables in which the number of missing data was higher than 5% with the specific intent to minimize the possible biases derived from the data missing.
Lines 110-127, state the names of the 12 centres, and number or ratio of data collected from each.
Response: We clarified the numerosity of each center with specified country and city. We added the following sentence: “In detail, the participating centres with the corresponding number of patients enrolled in the present study were: Padua (Italy; n=630), New York Columbia University/Weill Cornell Medical Center (USA; n=356), Brussels (Belgium; n=332), Hong Kong (SAR of People’s Republic of China; n=315), Innsbruck (Austria; n=313), New Delhi (India; n=270), Rome Sapienza University (Italy; n=248), Kyoto (Japan; n=230), Taiwan (Republic of China; n=200), Mainz (Germany; n=176), Kyushu (Japan; n=161), and Hangzhou Shulan Health Hospital/First Affiliated Hospital (People’s Republic of China; n=94).”
Method should state how the model was designed, how the model was validated, and how the upper tumour burden limit etc was identified/defined- you need to mirror issues in results with the methods.
Response: In Statistical analysis, we reported how the model was constructed and internally/externally validated. We also clarified the definition of upper limit for DS. We tried to mirror these issues in the Results part.
Results:
Line 143- Table 1- define Derivation set and Validation set under the table.
Response: We defined Derivation and Validation Sets under the table. We added the following sentence: “Derivation (n=2,318; 75.0%) and Validation Set (n=773; 25.0%) were obtained from the entire population. Block randomization was performed to maintain a similar representation of Western and Eastern centers in the two data sets.”
Table 1: The distribution between West and East re distribution of data not clear, need to be improved.
Response: West and East cases are clearly reported in the row “West/East”.
In the entire population: 1,912/1,179 (61.9/38.1)
In the Derivation Set: 1,437/881 (62.0/38.0)
In the Validation Set: 475/298 (61.4/38.6).
The P-value between Derivation and Validation Set is 0.8. We further underline that the creation of the Sets was done using a block randomization, with the intent to maintain a similar representation of Western and Eastern centers in the two data sets. In detail, a block randomization was performed to maintain a similar representation of Western and Eastern centres in the two data sets. The entire population was divided in twelve subgroups (“blocks”), each one corresponding to every centre participating in the study. Then, patients within each block were randomly assigned to the Derivation or the Validation Set using a casual number generator. We arbitrarily decided to use the 75.0% and the 25.0% of the entire population for constructing the two different sets.
Discussion:
Lines 304/305: give a reference
Response: We added some references reporting studies based on multicenter-based retrospective studies focused on HCC: “[4,10-12,14-18]”.
Line 304 “..with all the studies…” not correct. Why “all”.
Response: We removed the term “all”.
The language of the manuscript should be carefully reviewed.
Response: We revised the language of the study, trying to improve its quality.
Lines 315/316: ‘In fact…” should be clarified. Not clear,
Response: The model is based on radiological data available at first referral. Response after LRT is available only after this time point. Therefore, it is not possible to use this variable for constructing a model focused on “first referral only” data. We added the following sentence: “In fact, radiological response after LRT is available only at time of LT, making impossible to use this variable for constructing a model based only on “first referral” variables.”
Lines 345-347: ethical approval concerns, also we need ethical approval from other centres.
Response: We already reported that this is a retrospective study. We obtained the ethical approval from the leader center, namely Sapienza University of Rome. All the other centers obtained local approvals according to the ethical rules concerning retrospective studies. No modification of diagnostic strategies, therapies nor any other clinical approach were done for the present study. The study was also registered on ClinicalTrials obtaining an ID code.
We added the following sentence: “The Institutional Ethical and Scientific Review board of the coordinating center approved the study (Sapienza University of Rome, approval code 1214/2019). All the other centers obtained local approvals according to the ethical rules concerning retrospective studies. The study was registered at http://www.ClinicalTrials.gov (ID: NCT03595345).”
The only center coming from the People’s Republic of China, namely the Shulan Health Zhejiang University Hospital of Hangzhou was limited in its enrollment period (2015-2017) with the intent to avoid ethical issues.
Reviewer 3 Report
In current manuscript, LAI et al. performed an international retrospective study and developed a new model that identifies the upper limit of tumor burden for downstaging in hepatocellular cancer patients waiting for liver transplantation. This new model engages an intention-to-treat principle and can be used to predict the risk of post liver transplantation HCC recurrence. The study is properly designed and the manuscript was well written. I don't have any major critics. It would be great if more references can be added in the discussion to highlight the comparison with current models.
Author Response
Reviewer 3.
In current manuscript, LAI et al. performed an international retrospective study and developed a new model that identifies the upper limit of tumor burden for downstaging in hepatocellular cancer patients waiting for liver transplantation. This new model engages an intention-to-treat principle and can be used to predict the risk of post liver transplantation HCC recurrence. The study is properly designed and the manuscript was well written.
Response: We thank Reviewer#3 for the positive comments and for the opportunity to improve the quality of the study.
I don't have any major critics. It would be great if more references can be added in the discussion to highlight the comparison with current models.
Response: We thank Reviewer#3 for the comment. We added the following sentence with the intent to further highlight the role of WE-DS when compared with currently adopted models: “When the WE-DS Model was compared with the currently adopted MC and UNOS-DS Criteria [1,19] in terms of drop-out and post-LT HCC-related death, it was interesting to observe that only the WE-DS-OUT patients showed poor results. In detail, if the 5-year 30% survival cut-off for HCC-related death proposed by Mazzaferro was considered [18], only the WE-DS-OUT cases exceeded this value, further underlying the selection ability of the WE-DS model.”

Round 2
Reviewer 1 Report
Thank you for revising the manuscript, the overall quality and clarity have improved.
Reviewer 2 Report
The authors have improved the manuscript and addressed issues raised by the reviewer. It is suitable for publication.